# Neutral Electrolyzed Water in Chicken Breast—A Preservative Option in Poultry Industry

**DOI:** 10.3390/foods12101970

**Published:** 2023-05-12

**Authors:** Patricia J. Rosario-Pérez, Héctor E. Rodríguez-Sollano, Juan C. Ramírez-Orejel, Patricia Severiano-Pérez, José A. Cano-Buendía

**Affiliations:** 1Facultad de Medicina Veterinaria y Zootecnia, Department of Microbiology and Immunology, Universidad Nacional Autónoma de México (UNAM), Cuidad Universitaria, Mexico City 04510, Mexico; patomania@outlook.com (P.J.R.-P.); hector-eduardo_rs@comunidad.unam.mx (H.E.R.-S.); 2Facultad de Medicina Veterinaria y Zootecnia, Department of Animal Nutrition and Biochemistry, Universidad Nacional Autónoma de México (UNAM), Cuidad Universitaria, Mexico City 04510, Mexico; jrorejel@unam.mx; 3Facultad de Química, Department of Food and Biotechnology, Universidad Nacional Autónoma de México (UNAM), Cuidad Universitaria, Mexico City 04510, Mexico; pspmex1@quimica.unam.mx

**Keywords:** neutral electrolyzed water, chicken breast, *Escherichia coli*, *Salmonella* Typhimurium

## Abstract

Chicken is one of the most consumed meats in the world because it is an economical protein source with a low fat content. Its conservation is important to maintain safety along the cold chain. In the present study, the effect of Neutral Electrolyzed Water (NEW) at 55.73 ppm was evaluated on contaminated chicken meat with *Salmonella* Typhimurium and *Escherichia coli* O157:H7, which was stored in refrigerated conditions. The present study was carried out to check whether the application of NEW can help to preserve chicken breasts without affecting its sensory characteristics. Chicken quality was measured by analyzing physicochemical properties (pH, color, lactic acid, total volatile basic nitrogen, and thiobarbituric acid reactive substances content) after bactericidal intervention. This work includes a sensory study to determine if its use affects the organoleptic properties of the meat. The results showed that in the in vitro assay, NEW and NaClO, achieved bacterial count reductions of >6.27 and 5.14 Log10 CFU for *E. coli* and *Salmonella* Typhimurium, respectively, even though, in the in situ challenge, they showed a bacterial decrease of 1.2 and 0.33 Log10 CFU/chicken breast in contaminated chicken breasts with *E. coli* and *Salmonella* Typhimurium, respectively, after 8 days of storage, and NaClO treatment did not cause bacterial reduction. Nonetheless, NEW and NaClO did not cause lipid oxidation and nor did they affect lactic acid production, and they also slowed meat decomposition caused by biogenic amines. Sensory results showed that chicken breast characteristics like appearance, smell, and texture were not affected after NEW treatment, and obtained results showed that NEW could be used during chicken meat processing due to the chicken physicochemical stability. However, more studies are still needed.

## 1. Introduction

Poultry is part of the livestock industry, which transforms vegetable protein into animal protein [1,2,3]. The main chicken meat producers are the United States, Brazil, China, the European Union, and Russia [4]. These countries and supranational unions make the poultry industry a rapidly developing one. In developing countries, poultry farming plays an important role, as 8 out of 10 houses practice poultry farming (FAO). This is due, in part, to the rise in the price of chicken in stores, even though it is still the cheapest of all livestock meat [5,6], and its popularity is due, in part, to the fact that chicken is nutritious, is a versatile food in terms of how it is prepared, is healthier than pork or beef, and is also easier to digest [7,8].

Chicken meat handling during slaughter and the hygiene systems employed by operators throughout the production, distribution, and marketing chain significantly influence meat microbiology and physicochemical qualities. The main mechanisms involved in meat spoilage are microbial development, lipid oxidation, biogenic amine formation, and autolytic processes, but the speed with which these reactions occur depends on the intrinsic characteristics of the meat as well as extrinsic factors associated with meat handling [9]. Spoilage microbiota development is the main factor that causes alteration of fresh meat, and oxidation reactions have been recognized as the second most important factor, since they can modify the characteristics of smell, flavor, color, and texture, even during storage at low temperatures [10]. A significant portion of meat and meat sub-products (like giblets) are discarded due to the loss of freshness at different processing stages (like cleaning and inspection) of the production chain. Meat and meat products with signs of decomposition not only represent a decrease in the production chain but also constitute economic losses; they are a source of contamination, which can affect other food, facilities, and the environment [11], and if the chicken becomes contaminated, it would spoil rapidly and lose its microbial safety.

Chicken meat contamination by pathogenic bacteria like *Listeria monocytogenes* or *Salmonella* [12] is the main reason for the search for new alternatives to decontamination and preservation. Sodium hypochlorite and organic acids are used in decontamination processes, but they can be toxic and corrosive to stainless steel, or they could generate side effects on the organoleptic characteristics of the product [13]. The use of chlorine in the food industry has been banned in some European countries, such as Belgium, Denmark, and Germany [13,14,15,16]. Neutral Electrolyzed Water (NEW) is a germicide generated by a process of electrolysis of NaCl solution, which thus generates stable hypochlorous acid in a controlled way. NEW has already been used with favorable results in the disinfection of foods like strawberry, broccoli, squid, and pork, as well as surfaces and environments [17,18,19,20,21]; the method to obtain it is cheap, and, moreover, its use represents an ecological option because it does not involve handling potentially dangerous chemicals. Its mechanism of action against bacteria is attributed to the oxidation effect of the sulfhydryl (-SH) and the amino acid groups of the bacterial wall, which affects the process of bacterial respiration and nutrition [22,23,24,25].

The bactericidal effect of NEW in chicken breasts has been evaluated in different studies [26,27,28,29]. In the present work, the physicochemical properties of chicken breast were evaluated, and a sensory study of the meat treated with the evaluated solutions was carried out in order to study the impact of NEW by trained judges. To our knowledge, this is the first report with a sensory study in chicken breasts that have been treated with NEW.

## 2. Materials and Methods

### 2.1. Evaluated Solutions

The Neutral Electrolyzed Water (NEW) was provided by (Esteripharma Mexico S.A. de C.V., Estado de México, México). Its effect was compared with sodium hypochlorite (NaClO) (Quimica Rique, Cat. No. 7681-52-9, Estado de México, México), because it is the most common antimicrobial agent used in the poultry industry [30,31], and saline solution (SS) was used as a wash control solution.

### 2.2. Chemical Analysis of Solutions

Oxidoreduction potential (ORP), chlorine concentration and the pH of NEW, NaClO, and saline solution were all evaluated in the way described in a previous work [32]. The pH [33] and oxidoreduction potential (ORP) [34] were determined using a waterproof tester potentiometer (HANNA Combo HI 9812). Evaluated solutions (50 mL) were measured in triplicate. The iodometric method was used to evaluate the free chlorine content [35]; briefly, evaluated solution (50 mL) was taken in a 250 mL erlenmeyer flask with stirring. After, 100 μL of glacial acetic acid (CH_3_COOH) and 1 g of potassium iodide (KI) were added, and the sample was then shaken and placed in darkness for 5 min. Subsequently, the solution was titrated with sodium thiosulfate (Na_2_S_2_O_3_) 0.01 N until it became a yellow straw color. Finally, 1 mL of 0.5% starch solution was added and titrated with Na_2_S_2_O_3_ to a colorless solution [34].

### 2.3. In Vitro Microbicidal Assay

The microbiological assay was performed as previously described in [36]. Briefly, *Salmonella enterica* subsp. enterica serovar Typhimurium (ATCC-7251) and *Escherichia coli* O157:H7 (Migula) Castellani and Chalmers (ATCC 11229) were inoculated in 5 mL of TSA broth and incubated for 16 h at 37 °C with shaking. Bacterial concentrations were determined and adjusted to a concentration of 10^8^ CFU/mL, according to Mexican Standard 040 [37]. Bactericidal evaluations were performed in triplicate using the following solutions: NEW (55.73 ppm), NaClO (35 ppm), and SS, as reported in [38].

After, 1 mL of inoculum (10^8^ UFC/mL) was added to 9 mL of each solution (NEW, NaClO and SS) and vortexed for 30 s. Subsequently, 1 mL of the suspension was taken and neutralized with 9 mL of peptone water. Serial dilutions of pure cultures were performed using saline solution and 0.1 mL from each dilution was plated on a petri dish containing TSA (Cat. No. 210800, Bioxon. Estado de Mexico, Mexico) or MacConkey agar (Cat. No. 210900, Bioxon. Estado de Mexico, Mexico). The total viable bacterial count was determined.

### 2.4. Chicken Breast

Chicken breasts were collected from Centro de Enseñanza, Investigación y Extensión en Producción Avícola, UNAM, Tláhuac, Mexico City. Chicken breasts were stored in sterile polyethylene bags in containers with ice beds. Transportation lasted about 30 min to the laboratory.

### 2.5. Experimental Design

Chicken breasts were divided into three groups containing 15 chicken breasts each. The first group was contaminated with *Salmonella* Typhimurium and the second group with *Escherichia coli*. After this, chicken breasts were divided into three subgroups and treated with NEW, NaClO, and SS. Chicken breasts from the first and second groups were used for microbiological and color evaluation. Chickens from the third group were used for physicochemical analysis. Each chicken breast was kept individually in a plastic bag at 4 °C.

### 2.6. Chicken Contamination and Treatments

A bacterial suspension (1000 mL) of *Salmonella* Typhimurium or *Escherichia coli* was prepared (10^7^ CFU/mL); subsequently, 5 breasts were immersed according to their group for 10 min with constant movement [39,40,41]. After 1 min of drain, evaluated solutions (NEW, NaClO or SS) were sprayed using plastic spray bottles (26 sprays, equivalent to 13 mL) on contaminated chicken and left to act for 1 min.

### 2.7. Bacterial Survival Counts and Chicken Breast Storage

Each treated breast was placed in a sterile plastic bag with 200 mL of SS and hand rubbered for 1 min; after this, each chicken breast was transferred to another sterile bag and kept under refrigeration conditions for further analysis [39], such as the color and physicochemical properties of the breasts. Washed bacteria from each plastic bag were shaken for 20 s and serial dilutions were performed. Dilutions were plated in petri dishes with Salmonella-Shigella medium (Cat. No. 214400, Bioxon., Estado de Mexico, Mexico) or Tripticasa Soy Agar (TSA) (Cat. No. 210800, Bioxon., Estado de Mexico, Mexico) for samples contaminated with *Salmonella* Typhimurium or *E. coli*, respectively. Petri dishes were incubated at 37 °C for 16 h and survival bacterial numbers were reported as Log10 CFU/chicken breast (Log10 CFU/chb). This analysis was performed at day 1 and day 8 after treatment.

### 2.8. Color

CIEL*a*b* spectra was determined by using a colorimeter (Spectrophotometer CM- 600 d KONICA MINOLTA No. 21011486). Because it is important to detect the maximum change in the different elements included in the CIEL*a*b* spectra, the readings were taken on day 1 and day 12 after the treatments. Contaminated and treated chicken breasts were kept individually in plastic bags under refrigeration conditions. Chicken breasts were removed from plastic bags and readouts (from individual breast) from five random zones were taken, and mean values were calculated as described in [25,32,36]. After, the chicken breasts were returned to their plastic bags.

### 2.9. Chicken Breast pH

Chicken meat (5 g) was blended with 45 mL of distilled water for 1 min. The meat suspension was filtered using filter paper to remove the connective tissue and pH was measured in triplicate with a previously calibrated potentiometer.

### 2.10. Lactic Acid

Lactic acid content was calculated following the methodology reported by [32]. Filtered solution (25 mL) from chicken breast pH protocol was allocated in a beaker and three drops of 1% phenolphthalein were added. It was homogenized and titrated with a 0.085 N NaOH solution, until a pale pink color was obtained.

### 2.11. Total Volatile Basic Nitrogen

Determination of total volatile basic nitrogen (TVBN) was based on NMX-F-362-S-SCFI-2011 protocol [42] with some modifications. Chicken meat (5 g) was homogenized with distilled water and 0.4 g of magnesium oxide (MgO) in a 125 mL flask. It was stirred for 30 min and filtered. Afterwards, it was centrifuged for 15 min at 2000 rpm to precipitate magnesium oxide excess. 10 mL was then transferred to a 10 cm diameter petri dish (the edges of which were coated with petroleum jelly) and 2 mL of saturated sodium carbonate (Na_2_CO_3_) solution was added, mixed, and covered with the lid, to which 13 drops of saturated boric acid solution in glycerin were added previously. After 24-h incubation at room temperature, the drops from the lid were transferred to a 250 mL erlenmeyer flask with 60 mL of distilled water (pH = 5.1). Finally, 0.1 mL of 0.5% methyl red alcoholic solution and 0.1 mL of 0.4% bromocresol green alcoholic solution were added. It was titrated with 0.01 N hydrochloric acid until a pink coloration appeared. The following Formula (1) was used to calculate TVBN.
(1)$$\mathrm{TVBN}\;\left(\frac{\mathrm{mg}\;N}{100\;g\;\mathrm{chicken}}\right)=\left(\frac{\left(A-B\right)*\left(\frac{0.01\;\mathrm{mol}\;\mathrm{HCl}}{1000\;\mathrm{mL}\;\mathrm{HCl}}\right)*\left(\frac{1\;\mathrm{mol}\;N}{1\;\mathrm{mol}\;\mathrm{HCl}}\right)*\left(\frac{14.01\;g\;N}{1\;\mathrm{mol}\;N}\right)*\left(\frac{1000\;\mathrm{mg}\;N}{1\;g\;N}\right)}{25\;g\;\mathrm{chicken}}\right)*100$$
where: A − B (mL) = Corrected volume (HCl used vol.—blank vol.)

### 2.12. Thiobarbituric Acid Reactive Substances (TBARS)

Secondary compounds from lipid oxidation were quantified by the TBARS assay [9,43,44]. Briefly, chicken meat (5 g) with 10 mL of 5% trichloroacetic acid (Sigma Aldrich, Cat. No. T63399, St. Louis, MO, USA) was blended and then centrifuged at 10,000 rpm for 20 min. The supernatant was filtered using filter paper Whatman type 4. After this, 2 mL of the filtrate was transferred to a glass tube and then mixed with 2 mL of 80 mM 2-thiobarbituric acid (TBA). The reaction was carried out in a water bath at 94 °C for 30 min, after which the reaction was allowed to cool down in an ice bath. Absorbance was measured at 530 nm using a Perkin Elmer UV/VIS Lambda 2S spectrophotometer. The calibration curve was made following MDA quantification protocol [45,46]; briefly, MDA standard was serial diluted and readouts were taken with a spectrophotometer. The oxidation products were quantified as equivalents of malondialdehyde (mg MDA/kg of meat).

### 2.13. Instrumental Texture Analysis

Chicken breast texture characteristics were evaluated using the TA-XT2 texture analyzer (Stable Micro Systems, Godalming, UK). For their analysis, the breast samples were cooked in an Oster brand electric skillet at 130 °C for 4 min on each side for a total time of 8 min, and the internal temperature was monitored until reaching 75 °C centerpiece.

#### 2.13.1. Texture Profile Analysis (TPA)

Chicken breast texture characteristics were evaluated by instrumental texture profile analysis (TPA), following the methodology described by [47]. After cooking, the chicken breast was cut into 10 × 10 × 10 mm, as suggested in the work of Takei et al. [48].

The instrument parameters were set as follows: crosshead speed 1 mm/s, retention time 1 s, working distance 40% deformation and activation force 0.1 N [47]. A 50 mm diameter aluminum probe (code P50, Stable Micro Systems) was used. All samples were cut perpendicular to the longitudinal orientation of the muscle fiber according to the methodology described by [49].

TPA parameters including hardness (kg), fracturability (kg), elasticity (ratio), cohesiveness (ratio), chewiness, and adhesiveness (kg s) were calculated from the force-time curves recorded for each sample using the Texture Expert software version 1.0 [50,51].

#### 2.13.2. Warner–Bratzler (WB) Test

Chicken shear force characteristics were evaluated using the WB test, equipped with a 3 mm-thick Warner–Blatzer probe, following the methodology described in [48]. After cooking, the chicken breast was cut into 10 × 20 × 10 mm. Cooked chicken (eight rectangular cuboids from three chicken breasts) was used for evaluation. Instrument parameters were set as follows: crosshead speed 1 mm/s, working distance from 30 mm, and activation force of 0.2 N. All samples were cut perpendicular to the longitudinal orientation of the muscle fiber, according to the methodology described by U-Chupaj et. al [47]. Texture Expert software 1.0 (Stable Micro systems, Goldalming, UK) [51] was used to calculate the Firmness or WBS force (kg), and the cutting energy named Toughness (kg · s) [27].

### 2.14. Sensory Analysis

Chicken breast sensory evaluation was performed after 24 h of treatment on fresh non-contaminated chicken breast with evaluated solutions (NEW, NaClO and SS). For their analysis, the breast samples were cooked in an Oster brand electric skillet at 130 °C for 4 min on each side for a total time of 8 min, and the internal temperature was monitored until reaching 75 °C centerpiece, which, after that, was cut into 2 cm wide strips. The modified Flash Profile was used [47] because it is a descriptive fast methodology. Chicken cooked meat evaluation was performed by 21 trained judges, with an average age of 24. All judges were trained in descriptive methodology and belonged to the Sensory Evaluation Laboratory at the School of Chemistry, UNAM.

At the beginning, the generation of descriptors was completed. Each judge generated the largest number of sample attributes and grouped them by appearance, smell, taste, and texture. Subsequently, the attributes were reviewed collectively, and judges named them using a vocabulary that they considered pertinent to add to their individual list, as well as to eliminate the attributes that they did not wish to evaluate. Working with a group of trained judges in the descriptive methodology of various foods allowed the generation of a final list of attributes to be evaluated by all of the judges.

At this stage, relevant or intense attributes were evaluated. Terms that referred to the same attribute, such as softness, hardness, etc., were homologate. Finally, based on the definitive attribute lists, questionnaires were created that helped us to obtain a sample evaluation. FIZZ Ver. 2.3 program (Biosystemes, Couternon, France) was used for questionnaire design and evaluation. A structured scale was also used (values between 0–9, where zero was absence and 9 was the maximum intensity).

### 2.15. Statistical Analysis

Obtained data was evaluated by two-way ANOVA and Fisher’s least significant difference (LSD), *p* ≤ 0.05 using the Software GraphPad Prism Ver. 6.00 (GraphPad Software, La Jolla, CA, USA). For Flash profile, data statistical analysis was performed using a Generalized Procrustes Analysis (PGA), which was performed using the statistical software XLSTAT 2012, Addinsoft, version 10.0. Instrumental texture assessment results were evaluated using the Principal Component Analysis (PCA) methodology.

## 3. Results and Discussion

### 3.1. Chemical Analysis of Evaluated Solutions

Chemical properties of NEW, NaClO, and SS solutions are listed in Table 1. pH and ORP for NEW are within the expected intervals (pH 6.4–7.5 and 800–900 mV) [24]. The value of total chlorine was known, and in concentrations of 56–60 ppm of residual chlorine, a good antimicrobial effect has been obtained. These conditions provide greater stability and microbicidal effectiveness, and it is also known that residual chlorine concentrations of 56–60 ppm in electrolyzed water provide a good antimicrobial effect [52], whilst NEW had a slightly lower value than expected. NaClO solution showed a concentration that was subsequently adjusted without affecting its bactericidal activity. As we expected, saline solution showed a low ORP, and the chlorine concentration was below our detection methods, which we consider as a chlorine absence. Our results were similar to previous reports [29,53] where NEW properties were evaluated and reported, where the ORP was 1030 mV and had a pH of 6.7. The pH of NEW (used in this study) was similar, although the ORP was lower. The ORP of chlorine was similar to the reported values by Hernández-Pimentel and Guerra [29,53].

### 3.2. In Vitro Bactericidal Assay

Bactericidal activity results showed that NEW and NaClO (35 ppm) solutions are two effective germicides against *Escherichia coli* O157:H7 and *Salmonella* Typhimurium because both solutions achieved 6.27 Log10 CFU/mL and 5.14 Log10 CFU/mL significative reductions, respectively, when they were compared against SS (Table 2) after 30 s of contact. This germicide effect is greater; however, the detection system has a detection limit of 3 Log10 CFU/mL.

### 3.3. Bactericidal Assay on Chicken Breast

After confirming the bactericidal effect of the NEW and NaClO solutions, their effect on chicken meat contaminated with *Escherichia coli* O157:H7 or *Salmonella* Typhimurium was evaluated. Figure 1A shows the bacterial survival number obtained after 1 min and 8 days after treatment in contaminated chicken. During this time, meat was kept in sterile polyethylene bags under refrigeration conditions (4 °C). *E. coli* counts from first treatment showed no significant difference (*p* < 0.05) between groups, although obtained results at day 8 showed that NEW decreased 1.2 Log10 CFU/chb significantly and that NaClO did not cause any bacterial reduction. Sodium hypochlorite is not stable at a pH less than 7.5, while chicken meat has an initial pH close to 6.10 [54]. This could be a contributing factor to NaClO inactivation. When chicken was contaminated with *Salmonella* Typhimurium (Figure 1B), a decrease of 0.63 Log10 CFU/chb was observed when NEW was used, and those treated with NaClO showed a decrease of 0.59 Log10 CFU/chb. The decrease in *S.* Typhimurium at day 8 is only 0.2 Log10 CFU for those treated with NEW, but with NaClO, there was no decrease at all. It is important to consider that the chicken was immersed into an inoculum containing 10^7^ CFU/mL, which is a high number of bacteria that could affect the efficacy of the oxidizing solutions. If the chicken is contaminated during evisceration, too, the number of bacteria that could be in contact with chicken carcasses is not clear. Due to its physiological characteristics, *Salmonella* spp. can survive in refrigeration conditions at a pH between 3.8–9 with relatively simple nutritional needs. Given the nutritional characteristics of chicken meat and having an approximate pH of 7.1, this pathogen can survive in this matrix [55]. Previous works [56,57,58] where spray treatments were applied in chicken carcasses reported a reduction of bacterial numbers of 4 Log CFU/g and 9 Log CFU/g after 15 s or 3 min of duration of treatments with acid electrolyzed water (AEW) or NEW in combination with lactic acid.

Meat is a complex matrix due to its chemical compositions, and it is composed of water, proteins, amino acids, minerals, fats, fatty acids, vitamins, and other bioactive components, as well as small amounts of carbohydrates. Studies carried out by [40] showed that NEW can be inactivated in the presence of organic matter (proteins and fat mainly), and chicken meat has a high percentage of proteins and polyunsaturated fatty acids, so, possibly, the content of these nutrients causes NEW fast inactivation. Treatment was evaluated by immersion in contaminated breasts, where a decrease of 0.98 and 1.23 Log CFU was detected for *E. coli* and *Salmonella* Typhimurium, respectively, although we found no significant difference when S. Typhimurium was evaluated (Figure 2A,B). Another factor that influences the action of the evaluated solutions is the time of action of the solutions, since some authors recommend variable times according to the chlorine concentration of the disinfectant solution, so treatment times could be prolonged from 2 to 5 min [59].

There are some studies in which acidic and basic electrolyzed water (EW) have been used in immersion treatments [26,27,28,58], but these reports used AEW and the reduction counts were higher, and another difference was the time of treatments, which ranged from 3 min to 45 min. Another study [29] where chicken was immersed in NEW for 1.5 h reported a bacterial reduction of 1.2 log CFU/mL, and our results showed a similar trend, although statistical analysis reported no significant difference. Subsequently, as far as we know, this is the first report with sprayed NEW treated chicken, and a sensory study was carried out. Adaptive capacity responses of microorganisms, such as biofilm formation, may be another factor driving day 8 results. Biofilm is composed mostly of water (up to 97%) and exopolysaccharide, but it can also contain macromolecules such as proteins [60], which could participate in NEW inactivation. Some studies have shown that E. coli O157:H7 can develop biofilms as a result of the increased production of exopolysaccharides [61]. Furthermore, biofilm formation has been shown to provide increased bacterial resistance when they are exposed to hypochlorite solution, which is one of the most widely used disinfectants in the food industry [62,63].

One important factor in the use of water in poultry is the amount of water. In this study, we compared two application forms: spray and immersion. Immersion requires more water and we believe that spray treatments use less water and provide similar bactericidal effects, and so it is important to consider the amount of water when it is used in the chicken meat process [13]. However, the amount of used NEW could be less when it is applied by spray.

### 3.4. Physicochemical Properties of Chicken Breast

#### 3.4.1. pH

The pH values of the chicken breast developed after the treatments are indicators of the quality of the meat, it indicates the decomposition level, because after rigor-mortis stage and during storage and transport, muscle pH increases gradually, mainly due to microbial development and proteolysis; therefore, free amino acids and nitrogen bases generation contribute to increase pH. Chicken treated with SS and NaClO showed an increase in pH (Table 3), while NEW treatment, maintained significally pH values with fewer changes over time. This effect was reported as well in a previous work [29], where authors reported less pH values after 10 days of treatment. During the first two days after treatnment, chicken pH did not showed significant difference. This effect was reported by Kong et al. [64], when chicken breasts were thawed with slightly electrolyzed water. It has been repoted that a neutral pH helps to improve meat storage stability [65].

When pH is compared over time, we found that chicken meat treated with SS is significantly different at day 8 and day 12 and also when NaClO is used, pH is significantly different from day 3 to day 12, and the use of NEW helped to maintain pH values until day 12. As the days of storage of chicken meat in refrigeration passed, the pH values tended to increase, which can be attributed to the release of amines and carbonyl groups (from proteolytic reactions) [66] and, subsequently, to the decarboxylation of amino acids by certain genus of spoilage bacteria, including Escherichia coli O157:H7, which is how basic nitrogen compounds known as biogenic amines are formed. Data showed that NEW treatment maintained pH without changes over time, and this effect can be attributed to the fact that NEW acts on the sulfhydryl groups of proteins, which means that it may well be oxidized by *E. coli* O157:H7 enzymes that are responsible for amino acid decarboxylation. This effect, we should note, does not occur in treatments with SS and NaClO. Muscle pH has been associated with other characteristics of meat, including color, tenderness, water retention capacity, and microbial stability [67]. pH is thus related to shelf life to a large degree [68].

#### 3.4.2. Lactic Acid

When chickens are processed, the absence of oxygen and glucose in the muscle promote glycogen degradation by the anaerobic glycolisis pathway, and this pathway generates lactic acid (LA) production [69].

The percentage of lactic acid was measured to complement the pH results. Chicken treated with NEW and NaClO showed a slow and prolonged decrease (Table 4) of LA, and this was greater than that shown in the chicken breasts treated with SS. In chickens treated with SS, a sharp decrease was shown, and then an increase on days 4, 5, and 8. Lactic acid content was evaluated between the days, and the results showed that LA decreased by time when the chicken breasts were treated with SS, although NEW and NaClO treatments did not show significant differences by time, even though the values were decreasing with time. Having lower lactic acid values means that meat pH is higher, so it can be presumed that there may be a protein damage, and that the meat could be soft and exudative (or juicy) in texture due to the water retention properties of the meat being affected, since chicken meat is commonly consumed without any processing (unlike some other meat products). Soft texture and juiciness are, indeed, acceptable attributes in chicken meat.

#### 3.4.3. Total Volatile Basic Nitrogen Content (TVBN)

Biogenic amine formation is a consequence of amino acid degradation, which constitutes muscle protein, causing hydrolysis by both the endoenzymes and by bacterial amino decarboxylases that transform amino acids into amines. Due to bacterial metabolism, the biogenic amines that can be found correspond to cadaverine, putrescine, and agmatine, thereby enhancing the toxic action of histamine, and, thus, can serve as indicators of the beginning of a food’s decomposition. Chicken meat treated with NEW showed that there was less TVBN formation (Figure 3) during all 12 days, and chicken meat treated with SS showed the highest level during the storage. TVBN values were not significantly different when NEW and NaClO were used during the first 3 days, but at days 4, 8, and 12, chicken meat treated with NaClO showed higher values than the NEW treated group. These results infer that there was less decomposition when compared to both SS and NaClO treatments. These results showed that NEW has a preservation effect on chicken meat, and this can be attributed to its oxide reducing power, as this solution could be oxidizing sulfhydryl groups of amino-decarboxylase enzymes kept by the decomposition bacteria, including Escherichia coli O157:H7.

Chicken meat that was treated with NEW had significantly lower values of TVBN over the aforementioned time period. This effect was reported previously when slightly electrolyzed and slightly acid water were used [56] as well as NEW in immersed chicken breast [29]. TVBN limit content values in chicken meat have not been defined, but Tokur [70] reported 18 to 30 mg/100 mg in different types of fish; based on this report, chicken meat treated with NEW or SS kept values of 17.74 and 18.24 mg N/100 g, respectively, at day 12 after treatment, and chicken treated with NaClO, meanwhile, showed a 28.08 mg N/100 g at day 12. There is a relation between the TVBN data and the lactic acid content. Data showed that NEW helped to keep the TVBN values lower in treated chicken breasts, and there is an increment in the pH of chicken meat and a decrease in LA content; if chicken proteolysis exists by enzymes and bacteria, this will generate nitrogen compounds, pH will rise, and lactic acid will decrease, as happens when chicken breasts are treated with saline solution.

#### 3.4.4. TBARS

Lipid oxidation was quantified using the TBARS assay [71]. Unsaturated lipid compound oxidation affects the quality of meat, since this phenomenon occurs by chemical reactions, which, in many cases, are caused by environmental factors such as ultraviolet light and oxygen, or by microbial metabolism. Samples treated with NEW (Table 5) had decreasing MDA values. Data showed that there was no significant difference between groups, and this showed that although NEW is an oxidizing solution, it did not affect the fat present and, consequently, that there is no lipid rancidity present in the treated meat. However, when the storage times were compared, NEW kept constant levels of MDA significantly while SS and NaClO treatments showed significant differences during the storage time. This effect could have occurred because the oxidizing solutions had an initial effect on the chicken meat, which generated high values.

Obtained TBARS values were similar to the study by Kurk [72], where a high hydrostatic pressure technique was used as a bactericidal treatment. It has been reported that TBARS values of 1.006 mg/100g represent a great level of degradation and a loss of freshness [73], and, according to our results, the highest value was 0.591 mg/kg. It has also been reported that TBARS values 0.6–2.0 mg MDA/kg of meat can be detected by an inexperienced consumer panel as off flavors [74].

#### 3.4.5. Color

Total color change determination falls within the analysis of physical characteristics. This evaluation is important because the perceived color of the meat is one of the decisive factors in the acceptance or rejection of the product. L*a*b* color space, also referred to as CIEL*a*b*, is currently one of the most used color spaces to evaluate, as it enables the evaluation of color attributes, the identification of inconsistencies, and the precise expression of the results in numerical terms.

Luminosity depends substantially on the amount of free water that the meat has, and it is associated with the speed and magnitude of the pH change in the meat during the first 20 h postmortem. In general, the quality of the paler or brighter chicken was worse, which might indicate the presence of defects in the meat that are known as Pale Soft and Exudative (PSE) meat;. However, dark chicken can indicate defects known as Dry Firm and Dark (DFD), and in this type of meat, the pH does not decrease during maturation and, therefore, the meat becomes more susceptible to contaminating bacteria, causing a bad aroma that develops during the refrigeration period, not to mention a shorter shelf life [9,75]. Luminosity was slightly increased in the groups of chicken meat treated with NEW and NaClO (Figure 4A) at day 1, and the NEW group showed no change at day 12, but the group treated with NaClO showed a significant increase in its value at day 12 with respect to the SS group. The variation of these values may also be due to the type and age of the bird [54]. The increase of luminosity at day 1 with the chlorine treatment was similar to previous reports [76,77]. However, in another study, where chlorine concentration was higher (200 ppm), the luminosity did not experience a significant change [78].

Parameter a*, which is a coloration change from green to red; the results obtained showed that treatment with NEW and NaClO caused a decrease in the value (Figure 4B). This may be due to the effect of oxidizing solutions on myoglobin [64], since it is the main component responsible for the color of the meat, and this water soluble protein is constituted by a polypeptide chain called globin and a protoporphyrin ring in which an iron atom (Fe^2+^) is found. Its oxidation induces the formation of metmyoglobin, generating a process of discoloration or a browning of the fresh meat. In a study where ClO_2_ was evaluated, the a* pattern was the opposite of our results [76]. This could be because the main compound present in NEW is hypoclorus acid (HClO) and its oxidazion capacity is higher than ClO_2_. In another study, no change was reported when UV and chlorine treatment were performed [78].

Regarding parameter b* (Figure 4C), an immediate decrease in the values of the treated groups was detected at day 1 and day 12, which was the only significant change presented by the group treated with NaClO. Lipid oxidation has an important impact in the parameter b increase [79], and this might be related to the obtained values in the TBARS assay. Similar decreased b* values were reported in other works where chlorine dioxide was used [76], although a different study reported the opposite pattern when chicken was immersed in chlorine for 5 min [78]. NEW has a higher oxidation capacity, and this could be the reason why it affects the turn from yellow to blue more than anything else. 

To estimate the entire impact of the involved factors, ΔE value was calculated (Figure 5), where it was observed that both the NEW and the NaClO treatments do not change color at day 1 after treatment. This value showed that chicken color did not change after the treatment, although these values increased at day 12, when the greatest color change was detected by the group treated with SS, which was after the NEW group, and the group with only a minor change was the NaClO one. Kong et al. [64] compared different treatments during thawing of the chicken, and they concluded that the use of electrolyzed water did not change the color of the samples. However, another study reported that when chicken meat was immersed in NEW for 1.5 h, it did not show a significant ΔE value after 10 days of storage at 4 °C [29].

### 3.5. Texture

Texture results are shown in Figure 6, where component one explains 58.35% of the sample variability and component two explains 41.65% of it. The samples treated with NEW were characterized by being mainly cohesive and hard (hardness measured with the WBS probes), while the samples treated with SS were adhesive. The chicken treated with NaClO was characterized by being firm (measured with the WBS probe) and presenting attributes of instrumental TPA, such as less intensity, elasticity, chewiness, and hardness. Although more studies are required to evaluate the textural changes found in the chicken breast that had NaClO added, it has been reported that chicken meat with added NaClO during broiler pre-chilling reacts with the meat, producing dichloroacetic acid (DCAA) and tricholoacetic acid (TCAA)—reactions that are precursors found in chicken and water (carbohydrates, lipids, proteins, etc.) [80], which are macronutrients responsible for the texture of the meat. These types of reactions could contribute to changes in texture.

### 3.6. Sensory Analysis

Principal component analysis (Figure 7A), obtained from General Procrustes Analysis (GPA), showed that component one represents 57.93% of the sample variability and component two represent 42.07%. Chicken breast treated with SS correlated with components one and two, and it was characterized by its cooked appearance and its hard and chewy texture. Additionally, with chickens treated with NEW, the meat was positively correlated with component two and negatively correlated with component one because it presented a homogeneous look and a juicy texture, while the breast with NaClO presented spots in its appearance and an elastic, fibrous, and cohesive texture. Moreover, the aroma, taste, and remnant evaluation (Figure 7B) showed that component one explains 56.7% of the sample variability and component two explains 43.3% of that variability. Component one and component two were both positively correlated with the sample treated with SS, and it was characterized by the smell of fresh meat. On the other hand, though, the group treated with NEW showed a positive correlation with component one and a negative correlation with component two, and presented a salty and cooked chicken smell, a chicken flavor, and a salty remnant. Chicken treated with SS negatively correlated with the two components, and presented a roasted smell as well as a salty and fatty taste.

## 4. Conclusions

Chicken meat is one of the most important foods worldwide. This is the reason why it is important to produce safe food. The human population is growing as well as chicken demand. In developing countries, markets are far from production facilities, and storage and transportation are important targets to help to maintain chicken meat safety and quality. As an alternative to keeping meat, we compared the effect of NEW and NaClO on chicken breasts, and a sensory study was also conducted to evaluate whether the chicken was affected by the treatments. Electrolyzed water is considered environmentally friendly [81] because its reaction with organic material generates water and sodium chloride. In the evaluations where Neutral Electrolyzed Water was evaluated as well as NaClO, it was shown that both solutions had bactericidal activity against *Escherichia coli* O157:H7 and *Salmonella* Typhimurium, achieving a >5.14 Log_10_CFU/mL bacterial reduction in an in vitro study. NEW was shown to have a better bactericidal effect on contaminated chicken breasts than the NaClO solution, and the treatment can be sprayed. The use of NEW helps to stabilize the pH during storage time without affecting lactic acid concentration. Additionally, chicken breasts treated with NEW showed a lower formation of Total Volatile Basic Nitrogen as well as thiobarbituric acid reactive substances, such as malondialdehyde (MDA), when compared to NaClO-treated meat. The NEW treatment decreased the total color change of treated breasts after 12 days. In general, it was found that the chicken breast samples treated with NaClO 35 ppm and NEW presented similarities in their sensory characteristics, while the SS samples were different because the texture characteristics measured instrumentally presented similarity between the groups treated with SS and NaClO, and, thus, the NEW samples were different.

NEW is a solution that helps to slow down chicken meat decomposition without affecting the physicochemical properties of chicken breast or its sensory characteristics, such as appearance, smell, and texture. Only a slight salty taste was perceived in the sample treated with NEW.

This study showed promising data that suggest that NEW could be an alternative in the preservation of chicken breasts without affecting sensory characteristics, although its application needs more studies in the future. 

## Figures and Tables

**Figure 1 foods-12-01970-f001:**
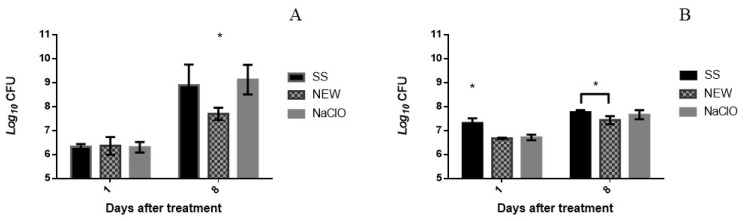
Survival numbers of *Escherichia coli* (**A**) and *Salmonella* Typhimurium (**B**) in contaminated chicken breast after days 1 and 8 of aspersion treatments. * Value with superscript is significantly different at *p* < 0.05.

**Figure 2 foods-12-01970-f002:**
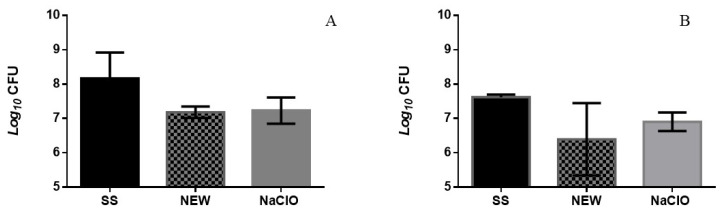
Survival numbers of Escherichia coli (**A**) and Salmonella Typhimurium (**B**) in chicken breast after 1 day of immersion treatments. *p* < 0.05.

**Figure 3 foods-12-01970-f003:**
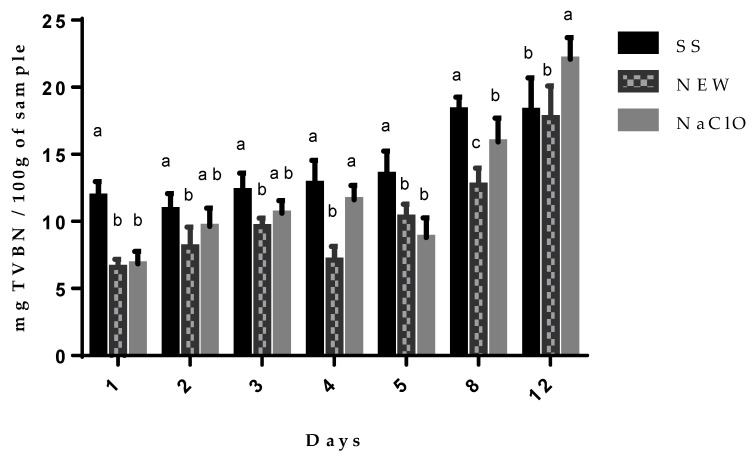
TVBN content on chicken breast after treatments. Values represent the means ± SD within days without a common superscript are significantly different. (*p* < 0.001).

**Figure 4 foods-12-01970-f004:**
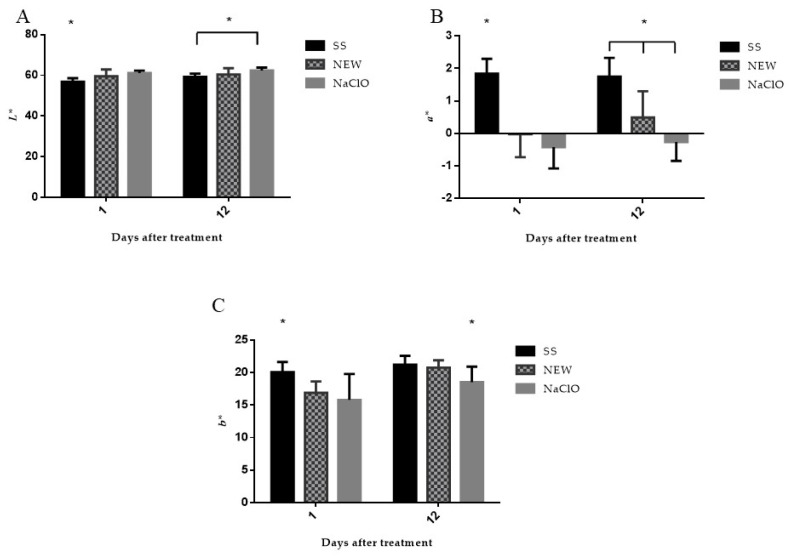
Differences in CIEL*a*b* color space of chicken breast in lightness (**A**), red/green (**B**), and yellow/blue (**C**) values after treatments. Values represent the means ± SD. * Value with superscript is significantly different at *p* < 0.001.

**Figure 5 foods-12-01970-f005:**
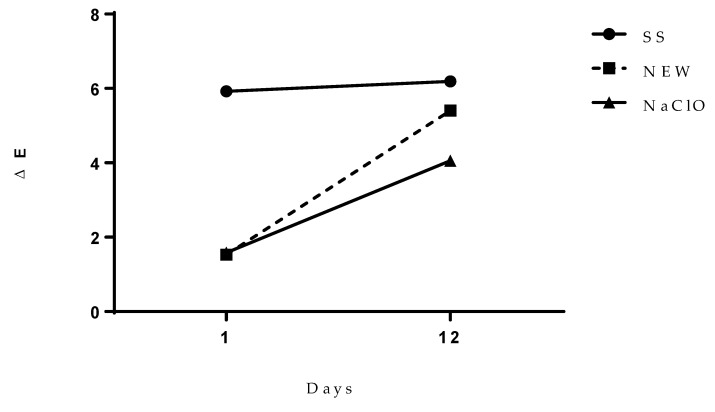
Overall color change (ΔE) of chicken breast after treatments.

**Figure 6 foods-12-01970-f006:**
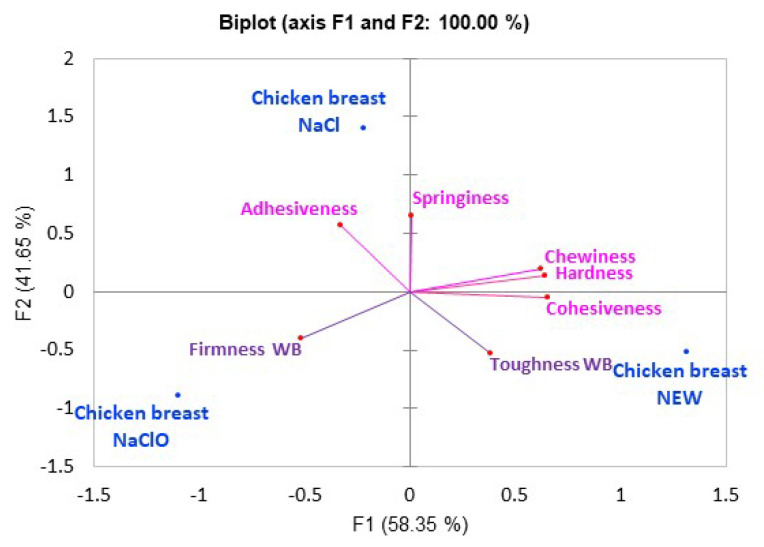
Principal component analysis (PCA in pink) of the instrumental texture profile analysis (TPA) and the measurement of hardness and firmness with the WB probe (purple) of the breast with different treatments.

**Figure 7 foods-12-01970-f007:**
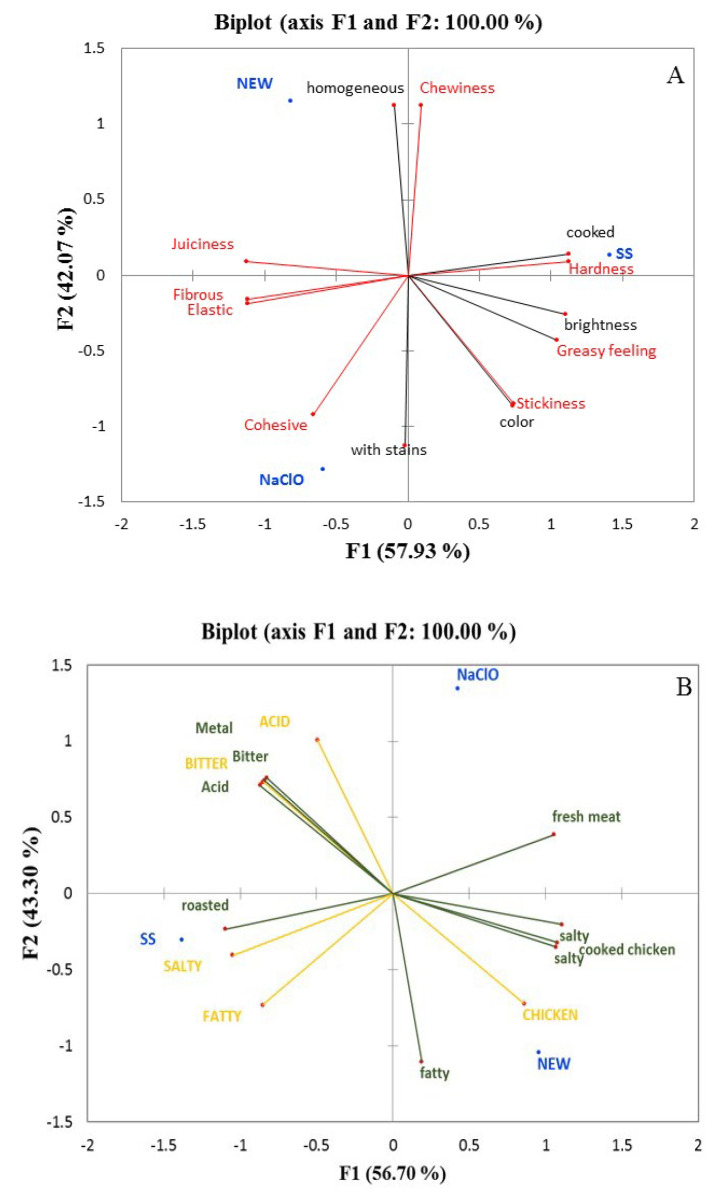
Principal component analysis (PCA) from a general procrustes analysis of the modified flash profile map, showing the relative sensory positioning of (**A**) appearance (green) and texture (red), and (**B**) smell (green), taste (yellow), and aftertaste (yellow) of the chicken breast treated with evaluated solutions. Rojo textura, verde olivo apariencia. Amarillo taste y verde olor.

**Table 1 foods-12-01970-t001:** Physicochemical characteristics of solutions.

	SS	NEW	NaClO
pH	5.86 ± 0.09 ^c^	6.70 ± 0.19 ^b^	7.58 ± 0.16 ^a^
ORP (mV)	365.67 ± 7.57 ^c^	895.67 ± 13.05 ^a^	692.33 ± 7.51 ^b^
Cl_2_ (ppm)	ND	55.73 ± 0.56 ^a^	35.02 ± 1.06 ^b^

Values represent the mean ± SD (*n* = 3). ^a–c^ Signifcant diference within each row (*p* < 0.0001). ND not detectable.

**Table 2 foods-12-01970-t002:** Bacterial survival (Log10 CFU/mL) after in vitro treatments.

	Treatment		
	SS	NEW	NaClO
*Escherichia coli*	9.27 ± 0.18 *	˂3.00	˂3.00
*Salmonella* Typhimurium	8.14 ± 0.03 *	˂3.00	˂3.00

Values represent the means ± SD. * Value with superscript is significantly different within each row at *p* < 0.0001.

**Table 3 foods-12-01970-t003:** pH of chicken breast.

	Treatment		
Day	SS	NEW	NaClO
1	6.10 ± 0.41 ^aC^	6.02 ± 0.16 ^aB^	6.10 ± 0.11 ^aB^
2	6.14 ± 0.33 ^aC^	6.11 ± 0.12 ^aB^	6.30 ± 0.33 ^aB^
3	6.29 ± 0.04 ^abC^	6.18 ± 0.11 ^bB^	6.57 ± 0.14 ^aA^
4	6.31 ± 0.07 ^abC^	6.22 ± 0.09 ^bB^	6.67 ± 0.07 ^aA^
5	6.27 ± 0.17 ^abC^	6.19 ± 0.03 ^bB^	6.63 ± 0.08 ^aA^
8	6.66 ± 0.63 ^abB^	6.43 ± 0.17 ^bB^	6.90 ± 0.09 ^aA^
12	7.30 ± 0.22 ^aA^	6.53 ± 0.05 ^bA^	7.10 ± 0.21 ^aA^

Values represent the means ± SD. ^a–b^ Significant difference within each row (*p* < 0.0001). ^A–C^ Significant difference within each column (*p* < 0.0001).

**Table 4 foods-12-01970-t004:** Lactic acid (g/100 g) in treated chicken breast.

	Treatment		
Day	SS	NEW	NaClO
1	0.161 ± 0.05 ^aA^	0.154 ± 0.03 ^aA^	0.127 ± 0.01 ^aA^
2	0.110 ± 0.01 ^aB^	0.136 ± 0.00 ^aA^	0.109 ± 0.02 ^aA^
3	0.129 ± 0.03 ^abA^	0.144 ± 0.01 ^aA^	0.100 ± 0.01 ^bA^
4	0.112 ± 0.01 ^aB^	0.135 ± 0.01 ^aA^	0.118 ± 0.01 ^aA^
5	0.136 ± 0.02 ^aA^	0.128 ± 0.00 ^aA^	0.114 ± 0.00 ^aA^
8	0.161 ± 0.04 ^aA^	0.120 ± 0.00 ^bA^	0.100 ± 0.02 ^bA^
12	0.107 ± 0.01 ^aB^	0.129 ± 0.00 ^aA^	0.108 ± 0.03 ^aA^

Values represent the means ± SD. ^a–b^ Significant difference within each row (*p* < 0.05). ^A–B^ Significant difference within each column (*p* < 0.05).

**Table 5 foods-12-01970-t005:** Lipid oxidation (expressed in TBARS, mg MDA/kg of meat) in treated chicken breast.

	Treatment		
Day	SS	NEW	NaClO
1	0.757 ± 0.35 ^aA^	0.591 ± 0.25 ^aA^	0.821 ± 0.33 ^aA^
2	0.377 ± 0.16 ^aB^	0.431 ± 0.22 ^aA^	0.365 ± 0.12 ^aB^
3	0.249 ± 0.03 ^aB^	0.360 ± 0.19 ^aA^	0.686 ± 0.25 ^bA^
4	0.297 ± 0.12 ^aB^	0.533 ± 0.14 ^aA^	0.417 ± 0.08 ^aB^
5	0.339 ± 0.16 ^aB^	0.344 ± 0.14 ^aA^	0.345 ± 0.12 ^aB^
8	0.129 ± 0.02 ^aB^	0.101 ± 0.04 ^aB^	0.347 ± 0.10 ^aB^
12	0.180 ± 0.03 ^aB^	0.304 ± 0.18 ^aA^	0.308 ± 0.12 ^aB^

Values represent the means ± SD. ^a–b^ Significant difference within each row (*p* < 0.05). ^A–B^ Significant difference within each column (*p* < 0.05).

## Data Availability

Data is contained within the article.

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
