# Peer review of "Neutral Electrolyzed Water in Chicken Breast—A Preservative Option in Poultry Industry"

_foods, 2023, doi:10.3390/foods12101970_

Round 1

Reviewer 1 Report

This study evaluated the effectiveness of Neutral Electrolyzed Water (NEW) on contaminated chicken meat with Salmonella enterica and Escherichia coli O157:H7, stored in refrigerated conditions. The aim was to determine if NEW application can help preserve and extend the shelf life of meat.

Novel value is missing as there are already several papers on the same topic:

https://doi.org/10.1016/j.japr.2020.04.001

Please find the attached file for more comments.

Author Response

We appreciate all your comments and we tried to corrected/modify all points that were asked to correct. We considered that the manuscript has a better presentation, and all the information was improved. We are attaching a document with all the modifications in blue.

Reviewer 2 Report

foods-2332296 Bactericidal efficacy of Neutral Electrolyzed Water in chicken breast. A disinfection and preservative option in poultry industry

The authors propose to study the effect of Neutral Electrolyzed Water on microbial, spoilage, and sensorial characteristics of chicken breast I found the objectives of the work interesting and worthy of being researched. In my opinion, the research was well conducted, and the material and methods are adequate to achieve the objectives proposed but need improvement in the description of the methodology used. Since the authors have not only studied the effect of NEW but also 2 other treatments SS (as a control) and NaClO, these must be referenced in the abstract and objectives.

So, I found the article interesting and the subject worthy to be published with major revision.

Comments:

Line 28 - ‘after 8 days of treatment.’ consider replacing with after 8 days of storage.

Line 39 – ‘Poultry is the most important livestock activity in the world because it is an industry that transforms vegetable protein into animal protein’ My question is, in the other livestock production such as cattle, sheep, goats, and pigs also not occur vegetal to animal protein?

Line 49 – ‘The main mechanisms involved in meat preservation’ consider replacing with The main mechanisms involved in meat spoilage’

Line 71 - ‘In the present work, the bactericidal effect of NEW in chicken breasts were evaluated, the physicochemical properties of chicken breast were evaluated, and a sensory study of the meat treated with the evaluated solutions was carried out.’ Rewrite this sentence.

Line 88 - ‘2.3 pH and ORP ‘Why this new section? Is it not the same as the 2.2?

briefly describe the methodology used

Line 97 – ‘Salmonella enterica subsp. Typhimurium’this is not correct, is ‘Salmonella enterica subsp. enterica serovar Typhimurium

Line 102, and 120, replace saline solution with SS

Line 115 – ‘Chicken breasts were divided in 6 groups containing five chicken breast each’ replace with ‘Chicken breasts were divided into 9 groups containing five chicken breasts each

Line 139 – ‘bacterial numbers were reported as log10CFU’ is log10CFU/mL suspension, in all the suspension or log10CFU/g meat ?? Please made corrections along all the text

Line 147 – ‘Read outs (from individual 147 breast) were from five random zones and all values were calculated as described in 148 [9,12,15].’ Which calculations do the authors state? The mean values? Please add to the text this information.

Line 150 - ‘2.10 Chicken breast acidity’ replace with ‘‘2.10 Chicken breast pH’  replace also in line 158

Line 182 – TBARS

Did the authors not make a standard curve? briefly indicate how it was done and how the results are expressed

Line 193 – ‘skillet at at 130°C’ remove the redundancy

Line 214- ‘2.15 Instrumental texture profile analysis (TPA)’

The title of this section is about a specific texture assessment method (TPA) but the authors later describe 2 methods the TPA and the WBSF. Please correct. There is no need to repeat procedures and equipment common to both methods. For example, in line 230, the meat cooking procedure and the equipment used are repeated. The sample size that is different in the 2 methods should be indicated when the specific method is described.

Regarding the TPA test:

. Was a WB probe used? there must be a mistake.

-‘working distance from 25 to 30 mm and activation force of 0.2 N. All samples were cut’ in TPA test the samples are not cut, which was a deformation percentage

- Which texture parameters are measured, indicate it with units

Regarding Warner-Bratzler shear force:

- 3 mm-thick Warner-Bratzler probe, is correct? 3 mm thickness?

- Line 240 the authors ‘cutting energy (Kg · sec), and area under the deformation force curve’ but in the results section these parameters do not appear and appear Firmeness, please make the appropriate corrections.

Line 252 ‘These results showed its greater stability and microbicidal effectiveness;’ How did the authors reach this conclusion?

Line 282 – ‘chicken meat has an initial pH close to 5.6-5.7’ I think that the values are normally higher, as those reported in line 290 (5,95), and as reported by authores (6.10), please made the adequate amendments.

Line 293 – ‘Salmonella enterica’ replace with Salmonella Typhimurium, see all the text and made the adequate amendments

Line 319 – ‘this  is the first report with immersion chicken and where the NEW treatment was carried out’

I don't think it's entirely true, here are 2 examples:

https://doi.org/10.1016/j.japr.2020.04.001

https://doi.org/10.1177/10820132209687

Line 354 – ‘Values showed a behavior of slow and prolonged decrease (Table 4), and it was greater than that shown in chicken breasts treated with NaClO.’ Which values the authors are to compare to NaClO?

Line 378 – ‘food. Chicken meat treated with NEW showed that there was less biogenic amine formation (Figure 3)’. The authors did not evaluate the content of biogenic amines but TVBN and in my opinion, TVBN does not only include biogenic amines but other compounds (ammonia) that is why where the authors wrote biogenic amines I think it is advisable to replace them with TVBN

Table 5 - What is the explanation for the values of TBARs decreasing over storage?

Line 403 – ‘Lab color space, also 403 referred to as CIELAB,’ replace Lab by L*a*b* (the CIELAB is the LAB but with an asterisk)

Line 410 – ‘as Sale Soft and Exudative (SSE)’ replace with ‘as Pale Soft and Exudative (PSE)

Line 421 – ‘This may be due to the oxidizing effect of solutions with myoglobin’; replace with ‘This may be due to the oxidizing effect of solutions on myoglobin;’

Line 488 – ‘Total Volatile Nitrogenous Bases’ replace with ‘Total volatile basic nitrogen’

Author Response

We are very thankful about all your suggestions. We present a new version of the manuscript which includes the asked modifications.

Thank you again for your help to improve our manuscript.

Reviewer 3 Report

The article "Bactericidal efficacy of Neutral Electrolyzed Water in chicken 2 breast. A disinfection and preservative option in poultry indus-3 try" is well-written and interesting. However, some details must be corrected.

Line 23-25: Badly written sentences. I suggest rewriting.

Line 39-40: Are you sure? Put a citation please.

In general, more references should be added throughout the manuscript.

Line 44-46: Put a citation please.

Line 71-73: Confusing sentence. I suggest rewriting.

Line 102: Typo.

2.6 Experimental design: Explain better please. I suggest adding a table.

2.10 pH: Another topic about pH?

2.13 2-Thiobarbituric acid reactive substances (TBARS): Typo. I'm not sure if this method is the most suitable.

Overall, the Results and Discussion topic is very poorly structured. Basically the results were exposed. There is little or no discussion, comparison and references.

Standardize the tables.

Line 294-295: Typo. Use an up-to-date reference. A 2002 reference is not adequate in this context.

3.4 Physicochemical properties: And about the discussion?

Line 326-338: There is no reference or discussion, comparison....

Line 449-450: Food or chicken?

Author Response

(The authors gave the same response as above.)

Reviewer 4 Report

Dear authors, the paper describes an interesting method that could be beneficial if it confirms the obtained results.

Unfortunately there are many shortcomings in the text that needs to be integrated and rewritten.

My comments follow.

ABSTRACT

Neutral Electrolyzed Water – quote the acronym

In general, do not put periods after theparagraph titles. Also check font and layout.

MATERIAL AND METHODS

L. 85: “Oxidoreduction potential (ORP) chlorine pH concentration,…”, commas are missing.

L. 79-81: the use of hypochlorite for carcasses decontamination is not authorized in all Country. Specify “were allowed or similar world”.

L. 85-93: put together in one paragraph.

L. 105: is diluted not neutralized. Better revise the sentence.

L. 105-106: how was the bacterial count performed? Which cultural medium was use?

L. 126-128:

L. 133: E.coli in italics

L. 137-139: Was the sterile bag removed before the measurement? If yes, how many minutes elapsed before the measurement?

L. 221: indicate which post hoc test was used. Moreover also sensory analysis were evaluated using the Principal Component Analysis (PCA) methodology.

RESULTS and DISCUSSION

L. 244 Pay attention to italics throughout the text

L. 246: How long was the treatment? 1 minute or 30 seconds ?

Table 2. Enter the unit of measure.

L. 322: WRC?

L. 326-338: insert bibliographic references and improve the discussion.

L. 341-356: insert bibliographic references and improve the discussion.

Figure 3. It is not clear where the differences are statistically significant. integrate figure and relative paragraph

L. 363-369: This evaluation is very important and needs to be better analysed. How do you explain that there are no statistically significant differences despite such different values? although some standard deviations are high I don't think it is enough to explain. Moreover improve the discussion and insert bibliographic references.

L. 374- 406 and Figure 4 and 5. There is no trace of statistical analysis, how came?

L. 415-430: insert bibliographic references and improve the discussion.

L. 437-443: insert bibliographic references and improve the discussion.

CONCLUSIONS

The evaluation is preliminary, a replicate of the experiment was not made and the data refer to a single sample/thesis/time. Better to say that it is a promising preliminary evaluation and that if the data are confirmed...

I also suggest adjusting the title.

Author Response

(The authors gave the same response as above.)

Round 2

Reviewer 2 Report

The authors have made the suggested changes, so it is accepted in its present form with a minor correction. Replace '2.14 Instrumental texture profile analysis (TPA)' with '2.14 Instrumental texture' because the texture profile analysis (TPA) test and the Warner-Bratzler (WB) test are both instrumental textute methods.

Author Response

Thank you for your comments. I will modify it according to your suggestion.

Reviewer 3 Report

Overall, the quality of the manuscript has been improved.

Author Response

Thank you for your help.

Reviewer 4 Report

Dear Authors,

the changes and additions made are satisfactory.

Author Response

Thank you for your help.